



# Local time dependence of auroral peak emission height and morphology

Noora Partamies[1,2], Daniel Whiter[3], Kirsti Kauristie[4], and Stefano Massetti[5]

[1]The University Centre in Svalbard (UNIS), Longyearbyen, Norway
[2]Birkeland Centre for Space Science, University of Bergen, Norway
[3]University of Southampton, UK
[4]Finnish Meteorological Institute, Helsinki, Finland
[5]INAF-IAPS, Institute for Space Astrophysics and Planetology, Rome, Italy

**Correspondence:** Noora Partamies (noora.partamies@unis.no)

**Abstract.** We investigate the bulk behaviour of auroral structures and peak emission height as a function of magnetic local time (MLT). These data are collected from the Fennoscandian Lapland and Svalbard latitudes from seven identical auroral all-sky cameras over about one solar cycle. The analysis focusses on green auroral emission, which is where the morphology is most clearly visible and the number of images is highest. The typical peak emission height of the green and blue aurora varies

from 110 km on the nightside to about 118 km in the morning MLT over the Lapland region. It stays systematically higher (at 118–120 km) at high latitudes (Svalbard) at nighttime and reaches 140 km at around magnetic noon. During high solar wind speed (above 500 km/s) nightside emission heights appear about 5 km lower than during slow solar wind speed (below 400 km/s). The sign of the Interplanetary Magnetic Field (IMF) has nearly no effect on the emission heights in the night sector, but northward IMF causes lower emission heights in the dawn over Lapland and during the noon hours over Svalbard. While

the former is interpreted as a change in the particle population within the field-of-view, the latter is rather due to the movement of the cusp location due to the IMF orientation. The morning sector heights also show a pronounced difference when previously detected pulsating aurora events have been excluded/included in the dataset, suggesting that this type of aurora is a dominant phenomenon in the morning and an important dissipation mechanism.

     The morphological evolution described by an increase of complex auroral structures in the midnight hours follows the aver-

age substorm occurrence. This effect is enhanced during stronger solar wind driving and during higher geomagnetic activity (as measured by auroral electrojet index, AL). During high solar wind speed, the high latitude auroral evolution shows particularly complex morphology, which is not limited to the nightside but rather only excluding the magnetic noon hours. An increase in the geomagnetic activity further enhances the structural complexity of the aurora in the morning sector.

## 1  Introduction

An overview of auroral evolution in magnetic local time (MLT) outlined by Akasofu (1976) and recently revisited by Knudsen et al. (2021) describes the evening sector to be most populated by auroral arcs, while the midnight and morning sectors are characterised by structurally most complex aurora: the midnight sector aurora driven by geomagnetic activity and the morning





sector aurora often appearing irregular and patchy. On the dayside the auroral emission equatorward of the polar cap boundary is predominantly green, while poleward of the polar cap boundary, on the newly opened field lines, the auroral emission is

primarily red (Frey et al., 2019).

In an earlier study of auroral occurrence Nevanlinna & Pulkkinen (2001) showed that the occurrence of all auroral observations peaks at around magnetic midnight. They analysed visually classified auroral types in image data from eight auroral all-sky cameras in 1973–1997. Six of the eight cameras were located in the Fennoscandian Lapland under the main auroral oval, while one camera was installed on Svalbard and another one in the sub-auroral zone in Finland. Their visual classification

of the auroral displays divided auroral structures into arcs, rayed aurora, patches and other auroral forms. The resulting MLT distribution showed a slight dusk-dominance for auroral arcs while both patches and other auroral forms were most frequently observed in the dawn sector. A similar MLT distribution of arcs, patches and omegas reported by Syrjäsuo and Donovan (2004) was based on 220,000 automatically classified images from a Canadian camera station Gillam in 1993–1998. The smallest class was omega bands consisting of 600 images, while the other two auroral classes included many thousand images. The highest

auroral arc occurrence was seen at around 21 MLT, the number of omega bands peaked at about 02:30 MLT, and patchy aurora occurred most frequently at around 6 MLT.

As shown by the above mentioned studies, the "known" auroral classes were small compared to the class "other auroral forms". To deal with the "unknown" auroral forms an automatically calculated structural index called auroral arciness was introduced by Partamies et al. (2014). Using this index they could include all auroral forms in a long-term analysis instead

of only the handful of commonly accepted morphological classes (i.e. an automated version of the Nevanlinna & Pulkkinen (2001) approach). They concluded that auroral arcs are always part of auroral displays in both hourly and annual averages. They found a positive correlation with the auroral arc occurrence and the solar wind driving, although the occurrence of more complex structures varied much more than that of arcs. The arc occurrence maximised in the dusk sector at about 17–20 MLT, and then steadily decreased towards the midnight and dawn, while the portion of more complex aurora increased in MLT after

about 21 MLT. This temporal evolution is in a good agreement with the previously mentioned studies of the MLT evolution in auroral displays. The arciness index was later used in a substorm study to examine the morphological evolution of the aurora during different phases of substorms (Partamies et al., 2015). Auroral arcs were also found in all substorm phases as a basic element of the aurora, but a clear morphological transition was observed to take place at substorm onset. Growth phases were found to be more arc-dominated while within minutes after the magnetic substorm onset the auroral display changed into more

complex auroral structures.

The morphology of the dayside aurora has been mainly studied by employing the auroral camera data from Svalbard in arctic Norway. For instance, Wang et al. (2010) classified image data from Ny-Ålesund station at 79° N in 2004–2009 into four morphological categories. The class of auroral arcs occupied one third their whole dataset. The arc occurrence increased to 60–70% towards 17 MLT in the dusk sector and the fewest arcs were observed around midday.


Measurements of auroral peak emission (or lower border) heights are sparse in the literature and primarily consists of case studies of different auroral arc-like structures, as recently summarised by Karlsson et al. (2020). The nightside and evening



arcs are found at 110–150 km heights based on either emission or electron density measurements (e.g. Sangalli et al., 2011; Aikio et al., 2018), while taller rayed arcs, high-latitude arcs or morning sector arcs have been observed to occur higher, at
about 130–170 km (e.g. Hallinan et al., 2001; Kintner et al., 2002).

To facilitate large-scale emission height studies an automatic method was developed for auroral images by Whiter et al. (2013). This method uses simultaneous auroral all-sky images from neighbouring camera stations with overlapping fields-of-view. The auroral emissions are projected onto both horizontal and field-aligned planes, and the height value is determined by comparing results between the different projections together with correlating the projections from the two stations. Applied to
Fennoscandian MIRACLE network camera data, the automatically calculated peak emission heights were used to study pulsating aurora (PsA) (Partamies et al., 2017) in contrast to other types of aurora preceding the pulsating structures. This study consisted of about 400 PsA events showing that the median peak emission height decreased by about 8 km at the beginning of PsA. The same conclusion was achieved for both green (557.7 nm) and blue (427.8 nm) auroral emission. This result provided evidence for PsA being associated with higher energy electron precipitation as compared to the types of aurora preceding PsA,
which was concluded to be substorm related auroral displays. Because PsA is a very common type of aurora in the morning sector, lasts for several hours and covers a large range of latitudes and local times, it is expected to be a dominant feature in the morning sector aurora causing lowering of the peak emission heights.

In this study, we analyse the magnetic local time (MLT) distribution of auroral arciness and peak emission heights calculated
for both Lapland and Svalbard image data. These datasets include over a decade of systematic imaging from both latitude regions. By employing the same methodology on similar data, we can directly compare the dayside and nightside aurora, as well as investigate the dawn-dusk symmetry. We examine the effects of solar wind driving and level of magnetic activity on auroral morphology and emission heights, and we further study the contribution of PsA to the auroral behaviour in the morning sector.

## 2   Data

The core dataset of this work consists of the Magnetometers – Ionospheric Radars – All-sky Cameras Large Experiment (MIRACLE) network all-sky camera (ASC) images (Sangalli et al., 2011). The network is operated and maintained by the Finnish Meteorological Institute (FMI) in collaboration with Sodankylä Geophysical Observatory (SGO) and INAF-IAPS Institute for Space Astrophysics and Planetology in Italy. We use images from the five Lapland stations (SOD, MUO, ABK,
KIL & KEV) during winter seasons in 1996–2006 and the two Svalbard stations (LYR & NAL) during winter seasons in 1999–2012 (for geographic locations and fields-of-view see Figure 1). All seven cameras were identical in design and largely run on the same imaging mode. These ASCs were equipped with narrow bandpass filters for auroral emissions at 630.0 nm, 557.7 nm and 427.8 nm, so called red, green and blue respectively. While red and blue images were taken once a minute, green images were captured once every 20 seconds whenever the Sun was more than 10° below the horizon. This imaging rate
resulted in about one million green images per station per winter season, which in Lapland spans from September until April





and in Svalbard from November until March. All images were automatically pruned (Syrjäsuo et al., 2001; Partamies et al., 2014), and our analysis only consists of pruned images, so those containing aurora.

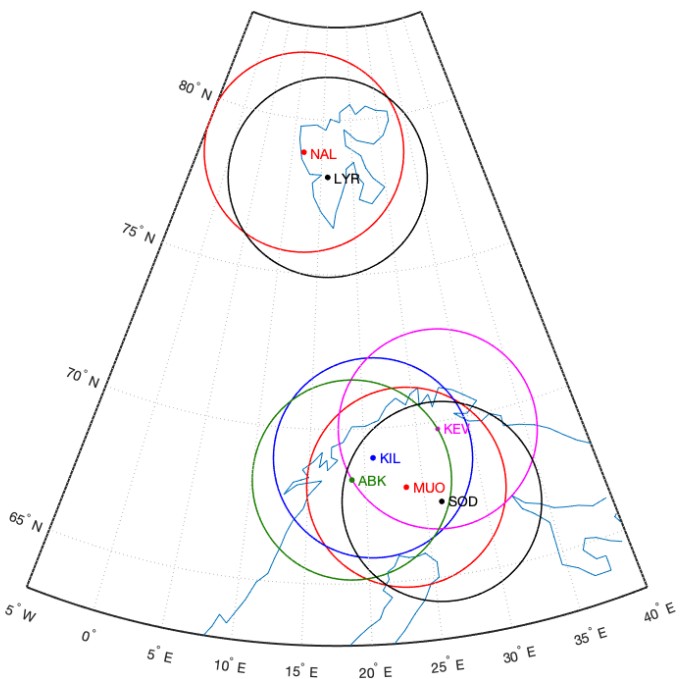

**Figure 1.** Geographic locations and fields-of-view of MIRACLE ASC stations. We use data from both the group of five cameras in Lapland (ABK – green, KIL – blue, MUO – red, KEV – pink, SOD – black) and the two cameras on Svalbard (NAL – red, LYR – black). The field-of-view circles are estimated at the height of 110 km and by excluding elevation angles below 10 degrees.

This study further utilises previously published data products of auroral peak emission heights (Whiter et al., 2013) and auroral arciness (Partamies et al., 2014). They both require overlapping fields-of-view (FoVs) of two neighbouring ASCs and

are thus called "paired data". Any height value from any camera pair in Lapland is considered as a Lapland auroral height in this study without specifying which ASC pair is in question, because the FoVs overlap so much that any station pair works well. Auroral peak emission heights are automatically determined by mapping the emission distributions to both a horizontal plane and many field-aligned planes (Whiter et al., 2013). When the mapped heights differ less than 20 km and the mapped emission projections show a correlation larger than 0.5, the heights are considered reliable and the heights from the field-aligned

projection are used for further analysis. For the Lapland ASC data in 1996–2006 the number of reliable peak emission height values was about 397,000, while the Svalbard ASC data in 1999–2012 gave about 83,000 reliable height values. One height value corresponds to the dominant auroral feature within the common FoV of one image pair. While this means ignoring the



peak emission height variations along auroral structures (e.g. Sangalli et al., 2011) this dataset provides a sufficient overview for large statistical studies, which is the aim here.

Auroral arciness is an index describing how arc-like an auroral display in an image is. This structural indexing is based on clustering of the brightest pixels in ASC images and fitting polynomials into the bright pixel clusters (Partamies et al., 2014). The arciness then describes the spread of the bright pixels around the polynomial fits. All arciness data we use is also paired data, so essentially data (image pairs) which also gave peak emission height values. As the pruning method (Syrjäsuo et al., 2001; Partamies et al., 2014), like any automatic classifier, is not perfect, there would always be some images of illuminated

clouds, for instance, which have been wrongly interpreted as aurora. Although there is strictly speaking no paired data requirement for arciness calculation, this selection guarantees that all analysed data actually contain aurora which is seen from two stations at the same time. The paired dataset adds up to about million images analysed for arciness and height over both Lapland and Svalbard. The arciness was described in detail and statistically analysed for the Lapland data by Partamies et al. (2014). As a Lapland arciness, we use the arciness data from the central station KIL in this study, which consists of a total of

about 357,000 values. The same procedure is repeated for the Svalbard image data, which provides the high-latitude and the dayside contribution to the auroral morphology. The number of Svalbard arciness data is based on Longyearbyen images with the total of about 183,000 arciness values.

    To determine the solar wind driving conditions we use the OMNIWeb data for solar wind speed and IMF. These data are

propagated to the magnetopause and have a temporal resolution of one minute. We divide the solar wind driving into *high* and *low* conditions, which are defined as solar wind speed less than 400 km/s or positive IMF $B_Z$ for low solar wind driving and solar wind speed larger than 500 km/s or negative IMF $B_Z$ for high solar wind driving.

    We further use the lower envelope curve of the auroral electrojet index (AL, Davis and Sugiura (1966)) to assess *high* and *low* geomagnetic activity conditions. The global AL index data are downloaded from the World Data Centre in Kyoto in one

minute time resolution. We define high geomagnetic activity as AL index values less than -300 nT and low geomagnetic activity as AL index values larger than -100 nT.

## 3   Results

### 3.1   MLT distribution of green and blue peak emission heights

Figures 2 and 3 show peak emission height data from the Lapland stations as a function of MLT (top panels) for green and

blue line images, respectively. At MIRACLE ASC stations the automatic imaging turns off when the sun is less 10 degrees below the horizon, so there are no images captured and no height estimates available in the middle of the day at any time of the year. The blue emission (Figure 3) is generally much fainter than the green one and blue images are taken once a minute while green images are taken three times a minute, so an order of magnitude less blue data has made it to this analysis. However, it is interesting to note that the median emission height evolution is very similar for both wavelengths without an obvious offset to

lower altitudes for the blue line. The typical emission height for the first bin in the evening (at 16–17 MLT) is about 110 km.



The median emission height then slowly increases up to about 117 km for green line at 2–3 MLT (Figure 2) and up to about 118 km for the blue line at 3–4 MLT (Figure 3). Thereafter the morning sector auroral emission height decreases to about 105 km at 7–8 MLT for green line and at 6–7 MLT for blue line. Beyond the mentioned morning MLT bins the number of data points (bottom panels) becomes so low that the median values may not be reliable any longer. The peak height of the blue line

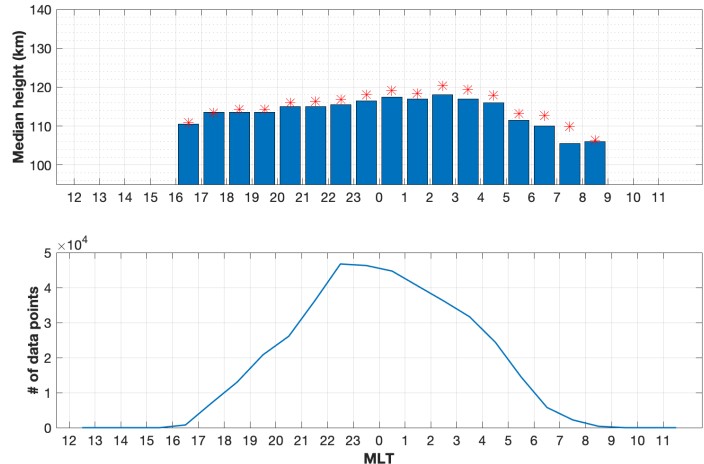

**Figure 2.** All peak emission height data from the green line images from Lapland camera stations averaged over one-hour MLT bins (top). The heights of the bars give the median value for each bin, while the red asterisks mark the mean heights. The empty bins are times when the daylight prohibits auroral imaging at Lapland latitudes. The number of all height data points for each MLT bin (bottom) demonstrates the nighttime bias in auroral imaging due to the distribution of the daylight. No seasonal selection is done here, but all height values from all years and all seasons are binned together.

is not considered further as it evolves similarly to the green one but in much lower numbers of data points.

Similarly, green peak emission height values from the Svalbard stations have been binned according to MLT hours in Figure 4. Similarly to the Lapland height data, most of the height data from the Svalbard ASCs are captured on the night side (lower panel). While the nighttime (at about 20–04 MLT) height values behave similarly to those in Figure 2, an increase in the emission heights dominates the dawn and dusk towards the MLT noon, where the maximum emission height of about

140 km occurs. The height decrease from the noon maximum towards the nightside minimum is steeper on the duskside (takes about 9 hours) than it is on the dawnside (lasts about 15 hours), illustrating the effect of more energetic particle precipitation in the morning sector on average.

### 3.2 Solar wind driving of peak emission height

We have examined the peak emission height changes with respect to the solar wind speed and the polarity of IMF $B_Z$. Higher

solar wind speed is typically related to stronger driving of the magnetopause reconnection, which may thus lead to more energetic particle precipitation into the atmosphere, in addition to higher fluxes of precipitating particles. Figure 5 shows



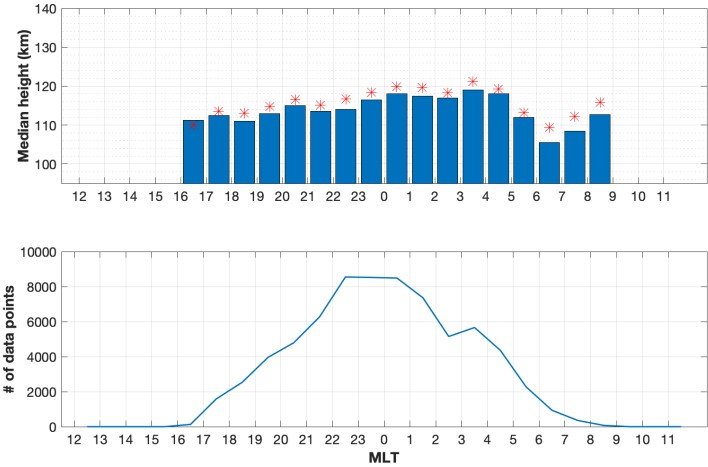

**Figure 3.** All peak emission height data from the blue line images from the Lapland camera stations averaged over one-hour MLT bins (top). The heights of the bars give the median value for each bin, while the red asterisks mark the average heights. The empty bins are times when the daylight prohibit auroral imaging at Lapland latitudes. The number of all height data points for each MLT bin (bottom) demonstrates the nighttime bias in auroral imaging due to the distribution of the daylight. No seasonal selection is done here, but all height values from all years and all seasons are binned together.

how this effect changes the median emission height as a function of MLT. The standard errors (dotted lines) are calculated as the standard deviation of the heights divided by the square root of the number of data points per bin. Their values are below 1.5 km on the nightside and up to the order of a few km on the dayside. The division into low ($V_{SW} < 400$ km/s) and

high ($V_{SW} > 500$ km/s) solar wind speed is of course somewhat arbitrary, but it is chosen so that each group still includes a statistically significant number of data points while agreeing with the typical solar wind speed being about 400 km/s, and speeds over 500 km/s often considered high. The Lapland height data (left panel) shows a systematic decrease from slow (red) to fast (blue) solar wind speed of about 3–10 km. The difference is largest during the early morning hours of 2–4 MLT and smallest in the dusk sector at 16–20 MLT. The nightside emission height values from Svalbard (right) are in a good agreement

with the Lapland ones (median around 120 km), while the dayside emission heights are 10–20 km higher for all conditions. At MLT noon about 5 km higher peak emission is seen during the fast solar wind speed (blue) than during the slow solar wind speed (red), while about 10 km higher emission heights are observed at around 8, 11 and 14–15 MLT during slow solar wind conditions as compared to fast solar wind driving. The median height at 11–12 MLT is about 25 km higher (161 km) for slow solar wind speed (red) than it is for fast solar wind speed (blue), but the slow solar wind value is based only on three data

points, which were captured on two different days, and is thus not representative for any bulk behaviour. It rather indicates that at this MLT the auroral activity is weak, in particular during slow solar wind. In general, more auroral data is available during fast solar wind conditions (blue) than during slow solar wind driving (red).

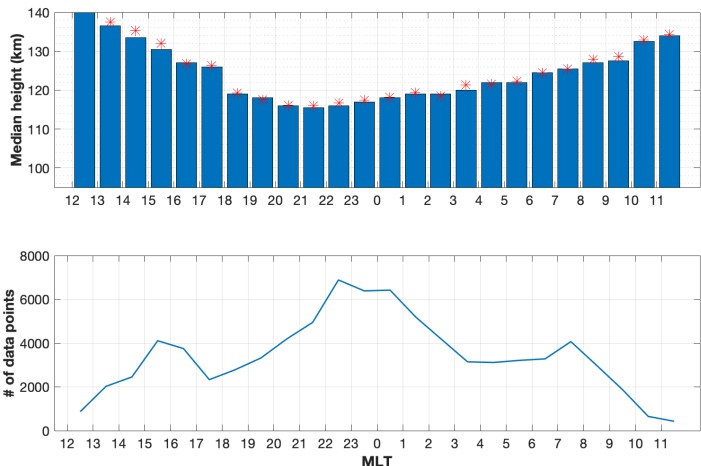

**Figure 4.** All peak emission height data from green line images from the Svalbard camera stations averaged over one-hour MLT bins (top). The heights of the bars give the median value for each bin, while the red asterisks mark the average heights. The number of all height data points for each MLT bin (bottom) demonstrates the nighttime bias in auroral imaging due to the distribution of the daylight. The median peak height at 12–13 MLT is 140 km and its mean value is 142 km. No seasonal selection is done here, but all height values from all years and all seasons are binned together.

Examining the effect of IMF $B_Z$ polarity (positive in red, negative in blue) in Figure 6 we find no notable difference due to the IMF polarity in the nightside emission heights in the Lapland data (left). Interestingly, in the dusk and dawn sector

the northward IMF leads to 2–5 km lower peak emission heights than the southward IMF. While the dusk sector difference is minor, the dawn sector at 5–7 MLT shows a more consistent decrease of emission heights from southward to northward IMF (of the order of 5–7 km). This likely reflects the shrinking of the auroral oval during northward IMF, which brings more energetic particle precipitation of diffuse aurora into the FoV of the Lapland camera stations in the late morning hours. There is some more variability in the median heights from bin to bin during the positive IMF $B_Z$ (red) than during the negative IMF

$B_Z$ (blue), because there is less aurora (fewer data points) during weak solar wind driving (slow speed & northward IMF) than during strong solar wind driving (fast speed & southward IMF). Changing the IMF threshold values does not change the fact that IMF polarity has no effect on the average auroral peak emission height on the nightside.

Over the high-latitude stations on Svalbard (right), however, southward IMF is associated with 3–5 km lower emission heights in most bins between 08 and 14 MLT. This difference is not uniform but consistent. At 18–02 MLT the median height

is around 120 km over Svalbard, just like it is over Lapland, while the dayside values vary between 130–140 km. At magnetic midday the southward IMF results in clearly higher peak emission height (by about 10 km) than the northward IMF, which is likely to be caused by the dayside reconnection moving the cusp region equatorward and into the Svalbard field-of-view. This replaces the more energetic boundary layer precipitation by less energetic cusp precipitation.





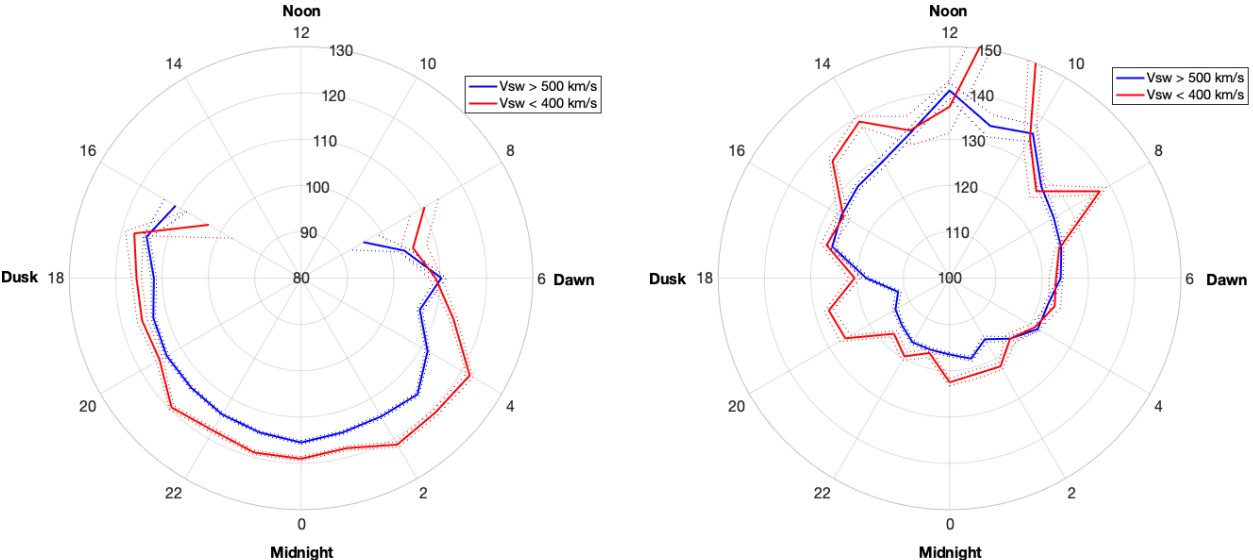

**Figure 5.** Peak emission height data from the green line images from Lapland camera stations (left) and Svalbard camera stations (right) during the times when the solar wind speed was less than or equal to 400 km/s (red) and during times when the solar wind speed exceeded 500 km/s (blue). Note that the radial height axis ranges are different: Lapland 80–130 km, Svalbard 100–150 km. The number of data points per bin varies from some tens up to several hundreds. The exception being the 11 MLT bin for slow solar wind speed, which only includes 3 data points. Standard errors (dotted lines) are calculated as standard deviation divided by the square root of the number of data points per bin. Some daytime bins have standard errors of several kilometres, but nightside errors are below 1.5 km.

A similar effect was seen when the peak emission height data was binned according to the auroral electrojet index values (distributions not shown). While the night MLT sector experiences lower peak emission heights during high magnetic activity (AL<-300 nT), the dayside emission height data suggests higher emission altitudes during high magnetic activity. The green auroral emission is typically less intense on the dayside than it is on the nightside, so the stronger solar wind driving or higher magnetic activity is required to enhance the occurrence rate of the green emission. We therefore have additionally visually checked all images included in any dayside bin with less than 100 data points to see that the data we include in the MLT distributions are meaningful.

### 3.3 Morphological changes in the aurora due to solar wind driving and magnetic activity

The energy from the solar wind driving is not only being converted to energy and flux of precipitating particles (as reflected by the peak emission height) but it also facilitates auroral dynamics. This is illustrated by Figure 7, which contains information on auroral morphology from the Lapland camera data (left column) and Svalbard camera data (right column). The results are divided into time periods of slow solar wind (top row) and fast solar wind (bottom row). The arciness data have been binned into Arcs ($A = 1$, blue) and other more complex auroral structures called Others ($A < 0.9$, red) to allow an assessment of





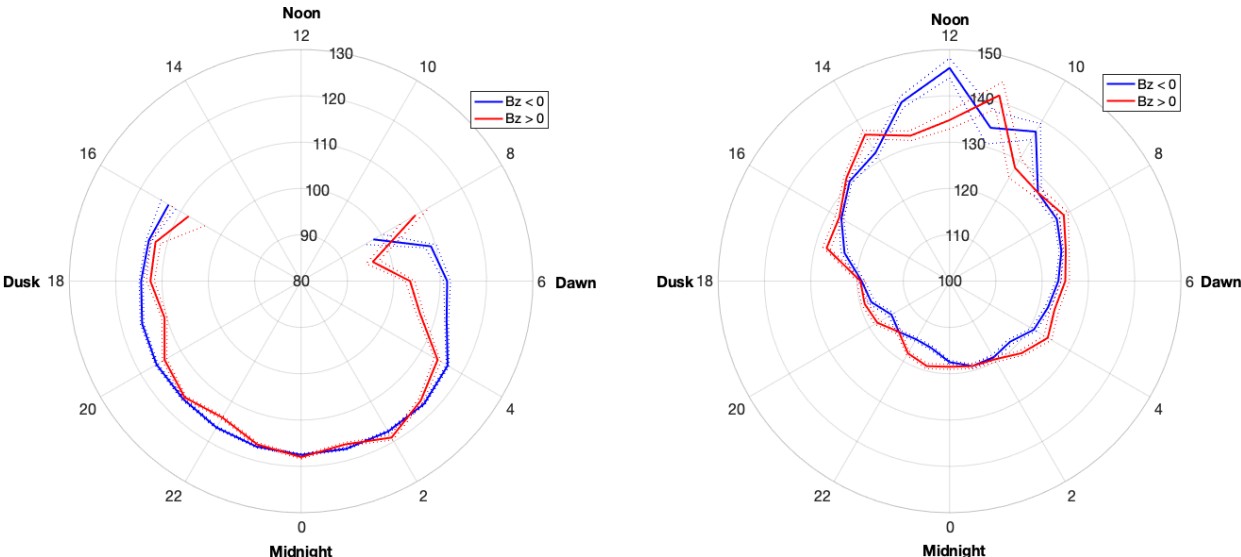

**Figure 6.** Peak emission height data from the green line images from Lapland camera stations (left) and Svalbard camera stations (right) during the times when the IMF $B_Z$ is positive (red) and during times when the IMF $B_Z$ is negative (blue). Note that the radial height axis ranges are different: Lapland 80–130 km, Svalbard 100–150 km. The number of data points per bin varies from some tens up to several hundreds. Standard errors (dotted lines) are mainly below 1.5 km.

structural changes as a function of MLT. The Lapland data show about double the number of complex aurora as compared to arcs in the midnight sector for both weak (top) and strong (bottom) solar wind driving. The difference between the two driving conditions is most obvious in the morning sector data. Between about 2–6 MLT the relative portion of complex morphology in the aurora is larger during periods of high solar wind speed (bottom) than during low solar wind speed (top). This difference is even larger over a much wider range of MLTs (about 16–06 MLT) for the Svalbard data (right column), where the relative number of arcs undergoes only small variations (mainly within a range of 200 data points), while the number of more complex structures fluctuates between 600 and 1300 data points.

In the structural distribution of aurora, the role of magnetic activity is pronounced as shown by Figure 8. Similarly to the previous figure we include the Arcs ($A = 1$, blue) and Others ($A < 0.9$, red) for Lapland (left column) and Svalbard data (right column) during periods on low (AL$> -100$ nT, top row) and high magnetic activity (AL$< -300$ nT, bottom row). The high magnetic activity (bottom row) drives a very distinct complexity in the structural evolution of the aurora as seen in the ASC images. Over Lapland, this is seen during the midnight and morning hours (about 22–06 MLT) with largest differences between Arcs and Others around and after MLT midnight. At the high-latitude stations on Svalbard (right column), the enhanced magnetic activity introduces a large population of complex auroral structures throughout the midnight sector (from about 18 MLT onwards) all the way to about 05 MLT in the morning.





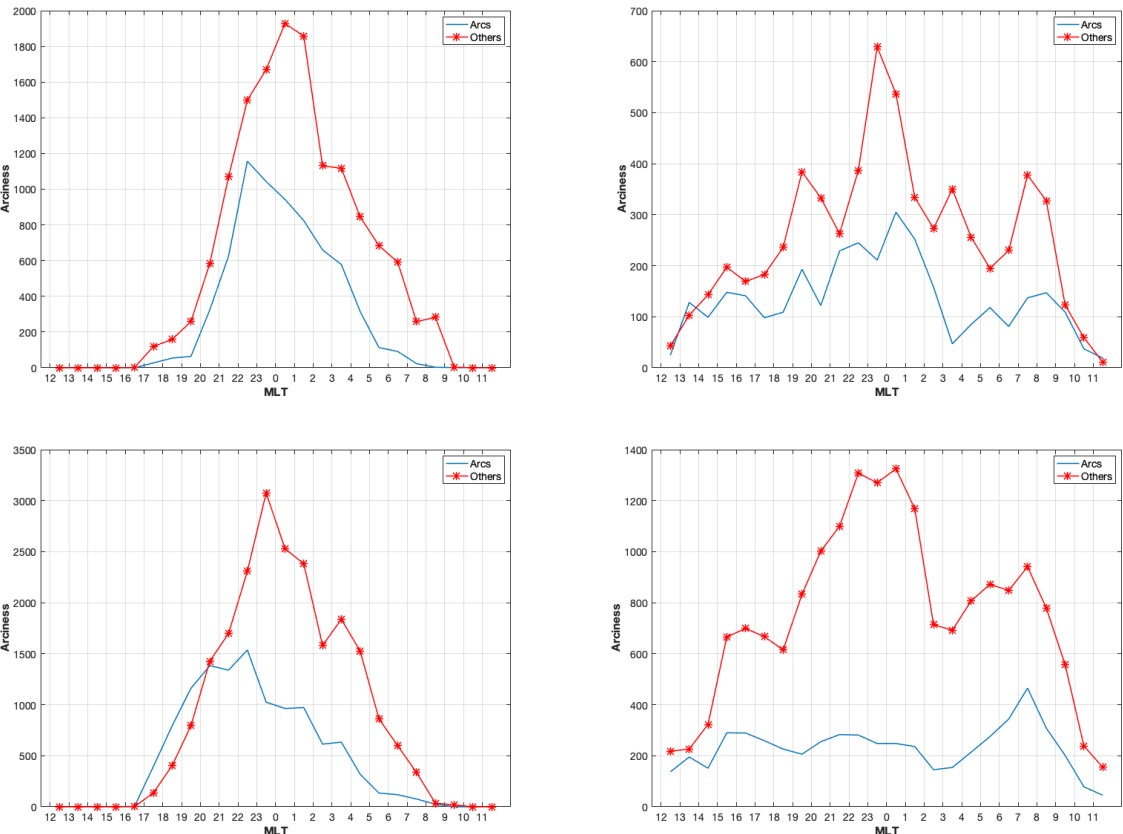

**Figure 7.** Numbers of analysed images with auroral Arcs (red) and Other more complex structures (A<0.9, blue) from the green line data from Lapland camera stations (left column) and Svalbard camera stations (right column) during the times when the solar wind speed is less than or equal to 400 km/s (top row) and during times when the solar wind speed is higher than 500 km/s (bottom row).

Much of the midnight activity can be attributed to substorms while the late morning sector complexity is more likely described by complex diffuse aurora, such as pulsating aurora. As it has previously been reported that pulsating aurora is very frequent in the morning sector, relates to high precipitation energies and evolves into complex patches, we examine the peak
emission height and the auroral arciness difference with and without data from the periods of pulsating aurora. We take the pulsating aurora time periods from Partamies et al. (2017) and exclude them from the Lapland data for arciness and emission heights (Figure 9). This is done without binning into the solar wind driving or magnetic conditions to keep the numbers of data points high. Excluding pulsating aurora in the full data set of heights removes about 52,000 images from 397,000 images in the analysis. Yet, this type of aurora is so concentrated into the morning MLT hours (over 50% probability estimated by Jones
et al. (2011)) that the removal of them increases the peak emission height by 3–7 km at 3–7 MLT (left panel). The effect of the PsA removal on the number of auroral arcs (right panel, blue & cyan curve) is negligible, while a detectable decrease is seen in





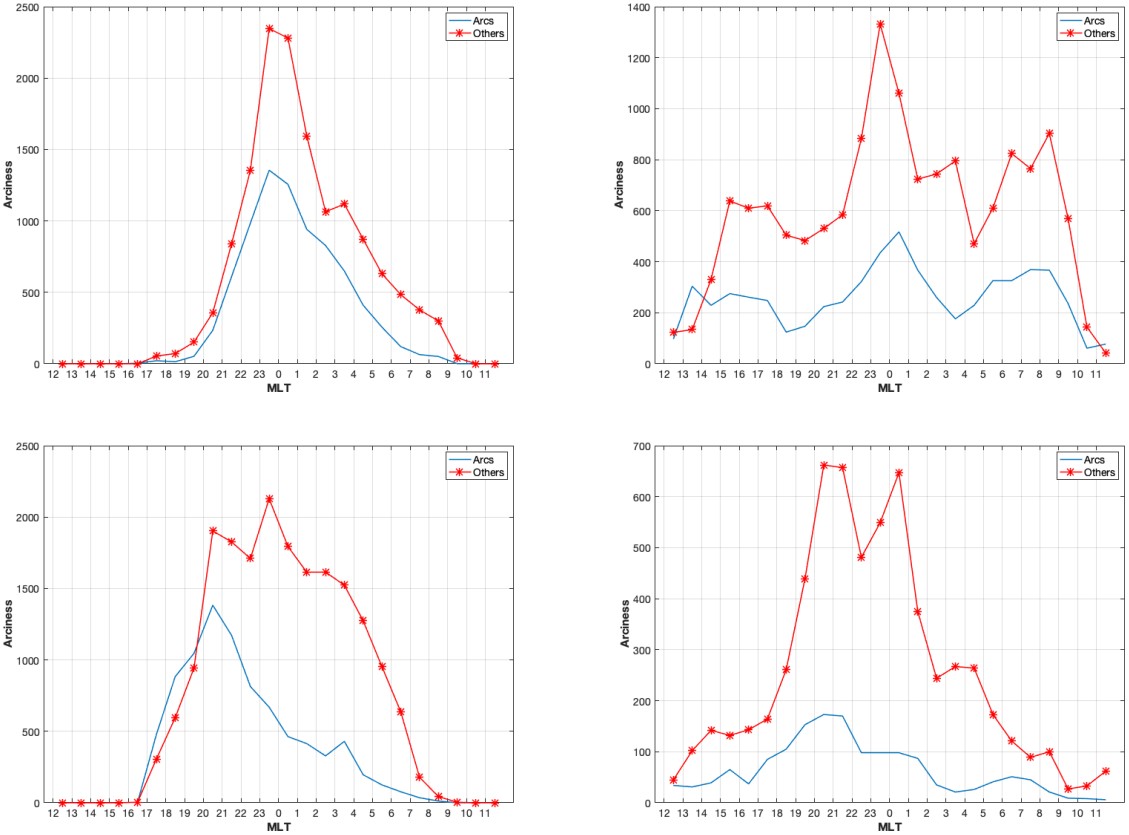

**Figure 8.** Numbers of auroral Arcs (red) and Other more complex structures (A<0.9, blue) from the green line images from Lapland camera stations (left column) and Svalbard camera stations (right column) during the times when the AL index is less negative than -100 nT (top row) and during times when the AL index is more negative than -300 nT (bottom row).

the number of Others (right panel, red & pink curve). This is indicative of PsA mainly occurring as irregularly shaped patches, which correspond to low arciness (values of about 0.4–0.7 (Partamies et al., 2017)). Without the time periods of PsA events, the morning side aurora is primarily left with more even diffuse aurora, tall rayed arcs, and other irregular forms.

## 4   Discussion

This study on the MLT dependence of auroral peak emission heights and auroral structures is, to our knowledge, the first large-scale study of its kind, employing about a half a million images from two different latitude regions over about a decade. Although the overall findings about the median peak emission height behaviour are not surprising, they provide a solid foundation for often assumed emission heights of 110–120 km for the nightside green aurora and 130–140 km (or higher) for the dayside green aurora. While the measurements of the peak emission heights tend to be limited to case studies (e.g. Sangalli et





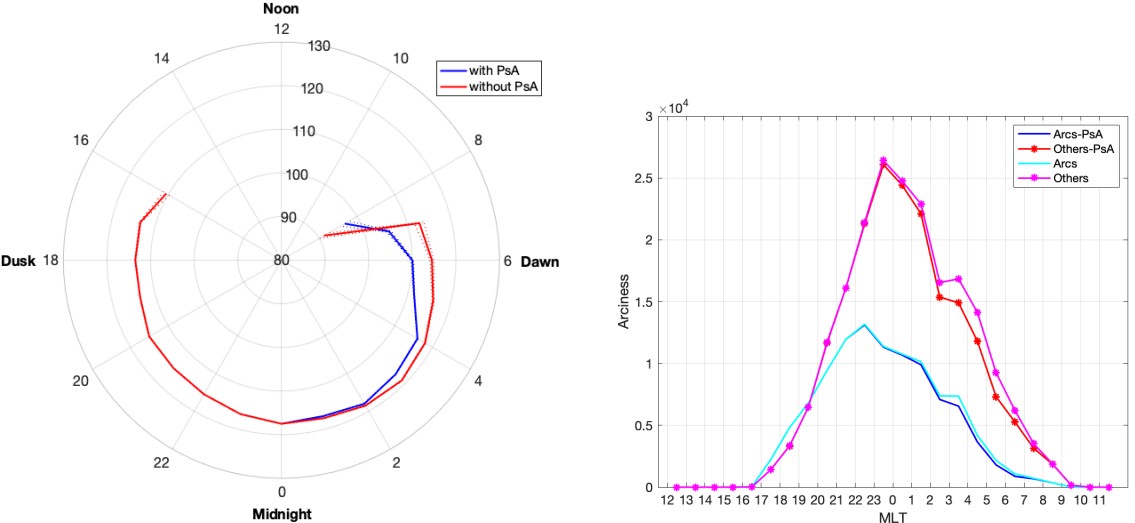

**Figure 9.** Green emission peak heights (left) as well as Arcs (blue & cyan, right panel) and Others (red & pink, right panel) with and without events of pulsating aurora as seen in the Lapland camera data. For heights the blue curve includes all data and the red curve excludes PsA. Standard error values (dotted lines) are below 1.5 km. For auroral arciness the dark colours (blue and red) show the distribution without the PsA events.

al., 2011), the studies of the particle precipitation energies (e.g. Newell et al., 1996; Motoba & Hirahara, 2016) tell a similar story, as 2–5 keV electron precipitation corresponds to a stopping height of about 110–120 km (Turunen et al., 2009).

The MLT distribution of peak emission heights behaves very similarly for green (Figure 2) and blue emission (Figure 3). As pointed out by Partamies et al. (2017): "The fact that the green and the blue emission heights undergo the same behavior

suggests that the green emission, at least in case of pulsating aurora, comes primarily from an energy interchange process between atomic oxygen and molecular nitrogen. This would make the peak emission heights of the green auroral emission also carry information on the precipitation energy." This is clearly demonstrated to be a valid process for all auroral structures in a companion study by Whiter et al. (2022).

Based on the long-term arciness evolution reported by Partamies et al. (2014), the solar wind driving conditions can bias the

distribution of auroral structures. This, however, is considered to be an insignificant bias in this study, since we only examine the relative portions of arcs and other more complex aurora, both of which are present in any solar wind conditions, in any MLT sector and in any substorm phase (Partamies et al., 2015). The outstanding auroral years of 2003 with very high average solar wind speed and magnetic activity, and 2008–2009 with extremely low solar wind driving and magnetic activity would cause the largest offsets in any auroral datasets. All these extreme years are included in our Svalbard dataset, while the lowest

activity years are beyond our Lapland dataset. Thus, the Lapland set of arciness may be biased towards more complex aurora as compared to the Svalbard arciness. In the nightside peak emission heights this may mean lower average peak emission heights for Lapland as compared to the Svalbard data. In these large amounts of data over a decade, however, the solar wind





speed distributions are the same for both time periods (1996–2006 for Lapland and 1999–2012 for Svalbard) within a few km/s.

While the fast solar wind is associated with lower peak emission height, in particular on the nightside aurora, the IMF polarity has its largest effects in the dawn, noon and dusk sector, leaving the nightside unaffected. In the dawn and noon sector, the northward IMF corresponds to lower peak emission heights as compared to the southward IMF, which we interpret as contraction of the auroral oval leading to a more energetic precipitation source region moving into the ASC field-of-view. This is particularly clear at noon where the cusp region of soft precipitation is narrow in latitude, just poleward of the more energetic

boundary layer particle sources, and sensitive to the IMF polarity (Newell et al., 2005; Frey et al., 2019). Nightside auroral oval is wider and the Lapland ASC FoVs are well within the band of discrete aurora. Nightside is thus less sensitive for the expansion and contraction of the auroral oval due to IMF changes with respect to our observation sites. In the dawn sector, where the diffuse aurora extends furthest equatorward during low solar wind driving (Newell & Wing, 2009), the contraction of the auroral oval may cause the typical emission heights to reduce. PsA events occur mainly under southward IMF but as about

one fourth of the events show pulsations lasting during northward IMF, these events can also contribute to the lowering heights during northward IMF. The dayside auroral emission is weaker than the nightside auroral emission, and the imaging season in the high-latitude polar night is also much shorter than the auroral season over Lapland. Thus, even for a decade-long time series of auroral image data, the number of pruned images (containing aurora) is limited when the data are binned according to other parametres, such as the solar wind speed and IMF here.


Structural changes in the mainland data are most likely due to substorm activity in the midnight sector. This observation is expected and in agreement with a general conclusion of an early long-term morphological investigation by Nevanlinna & Pulkkinen (2001), who concluded that the occurrence of active auroral forms follows the solar and geomagnetic activity better than the occurrence of auroral arcs does. Our results confirm that this is the case also in the statistical MLT distribution.

Furthermore, this finding is in a very good agreement with the arciness evolution over the substorm onset (Partamies et al., 2015), where the arc-dominated growth phase changes into complex aurora -dominated expansion phase. Occurrence of all auroral structures clearly peaks around the magnetic midnight during low solar wind driving, while during high solar wind driving the arc occurrence distribution widens towards the dusk and the distribution of more complex aurora extends further towards the dawn sector, again in agreement with Nevanlinna & Pulkkinen (2001). This enhanced structural complexity in

the morning sector has been known since Akasofu (1976) and is further summarised by Knudsen et al. (2021). Much larger variation over Svalbard likely relates to the expansion of the auroral oval during active times. While the Wang et al. (2010) study shows that one third of all dayside auroral structures are arcs, our statistics suggest that although one third is a representative number for high solar wind driving, arc occurrence reaches 50–70% during low solar wind driving (Figure 7). In agreement with Wang et al. (2010), our arciness dataset suggests that the most arc-dominated MLT sector is dusk at 17–20 MLT, although

our results refer to arcs being more frequent than other auroral forms in Lapland rather than in Svalbard. In the Lapland image data the portion of arcs exceeds that of other auroral structures when the solar wind driving or geomagnetic activity increases. This can be interpreted as a more frequent growth phase appearance in the FoVs of the Lapland ASCs in the pre-midnight





sector. The fewest arcs have previously been found at the MLT noon sector at 10:30–13:30 MLT (Qiu et al., 2016), while in our data the minimum arc occurrence appears particularly deep at 10–11 MLT. This time sector has also been called a "dayside

gap" due to low probability of accelerated particle precipitation (Newell et al., 1996) leading to an ultimate minimum in the occurrence of all aurora.

The presence of slow solar wind increases the auroral peak emission heights over Lapland strongest in the morning sector at 2–4 MLT (Figure 5, left panel). This is in agreement with the correlation between the occurrence of PsA and the solar wind speed (Partamies et al., 2017), as the increasing solar wind speed would drive higher PsA occurrence, and thus lead to higher

precipitation energies and lower peak emission heights. In this dataset the PsA has the highest occurrence rate at 2–4 MLT.

The height dive due to PsA in the morning sector seems to be the main reason for the dawn–dusk asymmetry in median heights. In fact, excluding the PsA events from the height distribution (red curve in Figure 9) removes the descending height evolution towards the late morning hours at about 2–7 MLT where PsA most frequently takes place (Tesema et al., 2020). What is left is rather a mild increase (2–3 km) in the median heights from midnight until about 3 MLT, after which a descending trend

is seen from about 120 km to about 115 km until 7 MLT. This suggests that other morning sector aurora does not have a major role in determining the auroral peak emission height, which is in agreement with earlier observations of emission heights being lower inside the pulsating aurora patches than outside the patches (e.g. Hosokawa & Ogawa, 2015). The studies by Hosokawa & Ogawa (2015) and Tesema et al. (2020) show that the most notable change in the morning sector precipitation takes place around 6 MLT after which the PsA related peak height of ionospheric electron density decreases from about 110–120 km down

to heights below 100 km (Hosokawa & Ogawa, 2015) as a result of soft precipitation ($< 10$ keV) decay while more energetic electron precipitation still persists (Tesema et al., 2020). As all auroral height data are analysed together in our study, we merely see a gentle decrease of heights towards the late morning hours, which nearly disappears when PsA events are removed. No PsA events have so far been investigated in Svalbard data, although auroral pulsations have been shown to extend to magnetic noon (Bland et al., 2019). While Lapland stations will never experience dark skies at noon, the Svalbard images can provide a

unique opportunity to examine the occurrence and properties of high-latitude PsA.

The increase of emission height by 3–7 km at 3–8 MLT when PsA events were excluded is in agreement with the 8 km decrease of height at the start of the PsA (Partamies et al., 2017). However, the height difference found in this study is a measure with respect to the background morning sector diffuse aurora rather than the substorm type discrete aurora where PsA often starts from. While in the previous study the start of the PsA event was associated with an average peak emission height

decrease from about 117 km to about 109 km, the all-inclusive median height in the morning sector shows a more gentle but a similar order of magnitude decrease of height from about 117 km at 4 MLT to about 110 km at 7 MLT, even though these data are not aligned for any event onsets. This, again, emphasises the role of PsA in the morning sector precipitation.

## 5    Conclusions

This study combines a decade of auroral all-sky image data from Svalbard and Fennoscandian Lapland to examine the magnetic

local time distribution of peak emission height and auroral morphology. The median peak emission height for the nightside

green aurora is about 120 km while for the dayside green aurora it is 130–140 km. Fast solar wind and enhanced geomagnetic activity drives lower peak emission heights on the nightside as compared to slow solar wind and magnetically calm conditions. The IMF polarity, on the other hand, has no appreciable effect on the nightside auroral emission heights, but changes the emission heights in the dawn, noon and dusk sectors, all of which is interpreted as the contracting auroral oval moving the

most poleward and softest particle precipitation regions poleward and outside the fields-of-view of the cameras.

The structural analysis is divided into auroral arcs and "Other" more complex aurora. It shows that the occurrence of more complex auroral forms follow the solar wind driving and geomagnetic activity more closely than the occurrence of auroral arcs, in agreement with earlier studies. The magnetic midnight sector is particularly prone to complex auroral structures due to substorm activity, but also the morning sector is dominated by more complex aurora. Magnetically quiet conditions or low

solar wind driving leads to up to 70% of the aurora being arcs, especially in the evening sector.

Excluding periods of known pulsating aurora from the green emission peak height dataset makes the MLT distribution of heights dawn-dusk symmetric by increasing the morning sector auroral heights by 3–7 km. This finding emphasises the role of PsA in the morning sector energy deposition.

*Data availability.* Global geomagnetic activity index data were obtained through Kyoto World Data Center (http://wdc.kugi.kyoto-u.ac.jp),

and solar wind data has been downloaded from the OMNIWeb database (https://omniweb.gsfc.nasa.gov). MIRACLE ASC quicklook data are available at (https://space.fmi.fi/MIRACLE/ASC/), full resolution image data can be requested from FMI (kirsti.kauristie@fmi.fi).

*Author contributions.* All authors have contributed to the discussion of the results and the writing of the paper.

*Competing interests.* The authors declare no conflicts of interest.

*Acknowledgements.* The work by NP is supported by the Norwegian Research Council (NRC) under CoE contracts 223252 and 287427.

DKW is supported by a NERC Independent Research Fellowship (NE/S015167/1). We thank the institutes who maintain the IMAGE Magnetometer Array: Tromsø Geophysical Observatory of UiT the Arctic University of Norway (Norway), Finnish Meteorological Institute (Finland), Institute of Geophysics Polish Academy of Sciences (Poland), GFZ German Research Centre for Geosciences (Germany), Geological Survey of Sweden (Sweden), Swedish Institute of Space Physics (Sweden), Sodankylä Geophysical Observatory of the University of Oulu (Finland), and Polar Geophysical Institute (Russia). Sodankylä Geophysical Observatory is acknowledged for the riometer data.

We thank S. Mäkinen, J. Mattanen, A. Ketola, and C.-F. Enell for maintaining MIRACLE camera network and data flow. The NAL all-sky camera is funded by the PNRA and the INAF-IAPS, its operation is also supported by the staff of the Dirigibile Italia Station and the Institute of Polar Sciences of CNR.



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
