# Peer review of "Local time dependence of auroral peak emission height and morphology"

_Annales Geophysicae, 2022_

## Author Response (AR1)

**Reply to comments by referee 1**:

This study made statistical analyses of several all-sky images in the Fennoscandian Lapland and Svalbard to estimate the aurora peak emission height. While I am thinking that explanation of the method can be improved by answering comments below, the results seem to be reliable and this study would be the first time to provide the MLT dependence from the large-scale data set. I think it is worthy of publication after revising.

*Thank you very much for helpful commentary.*

Since the authors have been working for a long time in this field, the scientific motivation and impact of this study should be addressed in the introduction, even briefly, I think. For example, optical data have been applied to estimation of physical parameters, such as precipitating electron energy and ionospheric conductivity, but these applications have not reached the satisfactory level. Then, what progress can be expected by applying knowledge of the peak emission height? In more simple manner, why would you like to know the peak height?

*Very true. The motivation for this type of studies is to understand the climatology of the auroral occurrence and behaviour in general. But it is the peak emission height of the aurora that would most closely relate to the precipitation energy of the electrons, and is thus interesting.*

*This has been phrased in the new version of the manuscript as: "Our motivation is to understand the*

*climatology of the auroral emission height and structure evolution. Furthermore, as the auroral peak emission height carries information on the energy of the precipitating electrons, the MLT distribution of emission heights in this large dataset will contribute to the knowledge of electron acceleration mechanisms as well.*
*"*

1. There are a few comments on the analysis method. Basically I agree with results from the analysis in this study, but there are several unclear points for me on the method. I hope that these comments will be helpful to make it clearer.

1-1. Section 2: I have a feeling that this section had better say more about the method to decide the peak altitude. Two methods have been presented in Whiter et al. (2013), method 1 (horizontal plane in Section 4.1) and method 2 (magnetic field lines in Section 4.2). Since the method 2 is a new one developed in Whiter et al. (2013), I am supposing that it has been applied on this study, but I am not sure because the text does not explicitly say. The text should address it.

*The field-aligned method ("method 2") is the newer one, and those values are used in this study (as well as all earlier studies using the same peak emission height dataset) when the values from both height methods agree. This is what the comparison of the different heights is and the maximum allowed difference of 20 km.*

*The text in section 2 says: "*When the height values from the two projections differ by less than 20 km, the heights from the field-aligned projection are used for further analysis.*"*

1-2. Section 2: According to the comparison in Whiter et al. (2013), the method 2 is less suitable than the method 1 for auroral arcs wider than 30 km or located along B in one of the images. Could you tell us how to deal with this issue in deriving the peak altitude?

*As the two methods are best suitable for different auroral structures, we require them to agree within 20 km. In most cases the heights from the two methods disagree much less than that (see next reply for more details). For the purpose of statistical studies, nothing more is done about the different method favouring different auroral conditions. If these values were used in case studies, the suitability of the different height methods could be evaluated more carefully.*

1-3. L98: "When the mapped heights differ less than 20 km and the mapped emission projections show a correlation larger than 0.5 …" I think that this study follows the criteria adopted in Section 4.2 of Whiter et al. (2013). Getting back to Whiter et al. (2013), I can't find the clear reason why these numbers (20 km and 0.5) were acceptable for the peak altitude determination. The text should tell us more about the background of the 20 km. Looking at the upper panel of Figure 2, the median/ mean peak altitude seems to vary within 105-120 km, which is shorter than 20 km. If the 20 km is "ambiguity" of individual estimation, I suspect the variation might be

dubious. On the other hand, the height difference might be usually smaller than 20 km. The text should mention the procedure of deriving the peak altitude in more detailed manner.

*These threshold values for good heights are indeed based on investigations of the height difference distributions in the original study by Whiter et al. (2013), where synthetic data was used for validation. The selection criteria are listed in that paper on page 135. In most real data cases the height difference from the two methods is less than 7km. For instance, changing this maximum difference from 20km to 10km is not affecting the results of this paper by more than a minimal reduction of the number of data points, while the mean and median values shown in this study do not change. The 20km difference in the heights between the two projections is implemented to exclude the outliers in cases when something went wrong rather than the ambiguity. Thus, the peak emission height variation of 105–120km is an actual detectable variability.*

*This has been clarified in the revised version as:* "While the maximum height difference between the two projections is implemented to exclude outliers, such as unfavourable location of the aurora low in the horizon, our statistical results are not sensitive to the choice of that value, since a height difference of 10 km gives the same results. 50% of height values used in this study differ by a maximum of 7 km."

1-4. Related to comment 1-3. Horizontal displacement of the mapped location due to the vertical difference is a

function of the elevation angle. It gets larger with smaller elevation angle (e.g., in case of the 20 km vertical difference, 0 km for the elevation angle of 90 degrees but over 100 km for 10 degrees). This means that a large vertical difference may not affect seriously in case of high elevation angles but the vertical difference should be a small value for the low elevation angle. This may suggest that the acceptable vertical difference should not be unique in the camera FOV. Accuracy of the peak height estimation may more rely on the horizontal difference rather than the vertical one, which may be related to the correlation coefficient. Could you tell us more about this point in the text?

*Not sure we understand this point correctly. Because the aurora is seen from two different elevation angles in two different images we are able to resolve the ambiguity between the horizontal and vertical displacements. Mapping of the images is not done in this study but just comparisons of height estimates by the two methods, which employ two different projections of the observed brightness distributions. If auroral structures with low elevation angles fail to produce comparable heights between the two methods the results are not included in the analysis.*

*We have changed the text to talk about projections rather than mapping to reduce the confusion.*

1-5. Caption of Figure 1 says that "… excluding angles below 10 degrees." Did this study analyze images by 10

degrees? If yes, that is fine. But if not, the smallest available angle should be applied in Figure 1.

*Yes, we exclude pixels below 10 degrees of elevation angles as the pixels become too large. An explicit statement about this has been added to the text: "The full all-sky FoV is reduced to exclude elevation angles below 10 degrees.", so it is no longer only mentioned in the figure caption but also in the main text.*

2. Figures 2 and 3: I have been supposing that the peak height of the green line is higher on average than that of the blue line, but these two figures do not clearly show the difference through the night. However, for the 6-7 MLT bin, the blue-line height may be significantly lower than the green one. Could you give us comments in the text on (1) no clear height difference in between for most of the night hours and (2) the difference for the 6-7 MLT bin?

*Perhaps for historical reasons, many scientists assume that the green emission peak height is higher than that of the blue emission. Statistically, that is not the typical case. A companion paper by Whiter et al. 2022 (submitted as a discussion paper, https://doi.org/10.5194/angeo-2022-23) contains a statistical study of the height difference between green and blue emission and shows that the two emissions peak at about the same heights, blue even higher than green in certain conditions. This thought is rephrased in the discussion section.*

*We do not have a good explanation for the blue-to-green height anomaly at 6-7 MLT, but the fact that for the blue peak emission height values in that bin the median and mean values are quite different hints to the direction that there is simply very large height value variability in that MLT bin.*

3. L142-: "Similarly to the Lapland height data, most of the height data from the Svalbard ASCs are …" Figure 4 shows the time series of the emission height of the green line in Svalbard, and the text says its similarity with the Lapland result. How about the result of the blue line in Svalbard?

*This statement only compares the data numbers between day and nighttime over Svalbard and Lapland. But it is true that the Svalbard blue peak emission heights are similarly distributed than the Svalbard green emission heights. As there is no notable difference (just like there is no on the Lapland data), we have not included the histogram for the Svalbard blue peak emission heights here.*

4. L168: "… in Figure 6 we find no notable difference …" As the text says after this part, there is a consistent decrease of the emission height in the dawn sector of the Lapland. So I have a feeling of wrongness on "no notable difference."

*We have phrased this more precisely to say that no notable difference refers to the nightside region at 20–04 MLT, and that there is a height decrease towards the dawn after 4 MLT.*

5. L179: "… in most bins between 08 and 14 MLT" This should be revised as "in most bins from 14 to 08 MLT", I wonder.

*Yes indeed. Fixed.*

6. Figure 6: The text does not say effects of small data points in the noon sector, which is different from Figure 5. I am thinking that the data point is good enough to make reliable statistical analysis for all time bins, but let me make sure if it is true.

*As the figure caption says, the number of data points per bin varies from tens to several hundreds, which we consider statistically sound. The lowest number of data points is found at 11 MLT, and it is about 50 for both IMF polarities. Correspondingly the standard errors in that bin are larger than elsewhere.*

7. Figure 7: Title of the vertical axis is "Arciness" but the figure caption says "numbers of analyzed images". I am thinking the figure caption is correct.

*Yes, very true. This should be the number of data points and will be fixed in the revised version of the figure.*

8. Figure 7: All four panels present a peak in the midnight and decreasing toward morning and afternoon. This pattern relies on number of the dark night (see Figures 2, 3 and 4) along with the event number. So the ratio may be better to capture the feature. Same comments for other later figures.

*True, the variability in the number of data points is*

*responsible for the decay towards dawn and dusk. The ratio between the number of Arcs and the number of Others essentially shows the arcs being the strongly dominant feature in the pre-midnight MLT and that that dominance decays towards the dawn. That behaviour is so strong, and the number of Arcs is so high, that it makes it harder to see the changes as compared to the absolute value plots we currently have. We therefore prefer showing the absolute values, but will add an explicit comment on the fact that the decay towards dawn and dusk relates to the decay in the number of data points. The example ratio figure here corresponds to Figure 9b in the manuscript.*

[Figure]

9. Figure 7. In the right panels for the Lapland result,

from 12 to 15 MLT (or may be 16 MLT) and from 10 to 11 MLT, markers are plotted at 0. It means that there is no event, but in reality, no measurement, I wonder. To make it clear, no marker is better, I think. Same comments for other later figures.

*The zero values in the left panels for Lapland do indeed mean that there are no data. These markers will be excluded.*

**Reply to comments by referee 2**:

*Thank you very much for helpful comments.*

Motivation: I agree with the other reviewer that the authors could be a bit more explicit on the motivation of the study in the introduction. What useful physical information can be derived from the emission height?

*The motivation for this type of studies is to understand the climatology of the auroral occurrence and behaviour in general. But it is the peak emission height of the aurora that would most closely relate to the precipitation energy of the electrons, and is thus interesting.*

*This has been phrased in the new version of the manuscript as: "Our motivation is to understand the climatology of the auroral emission height and structure evolution. Furthermore, as the auroral peak emission height carries information on the energy of the precipitating electrons, the MLT distribution of emission heights in this large dataset will contribute to the knowledge of electron energy distribution as well. Furthermore, as the auroral peak emission height carries information on the energy of the precipitating electrons, the MLT distribution of emission heights in this large dataset will contribute to the knowledge of electron acceleration mechanisms as well."*

Time averaging: If I am not mistaken, the statistical connection between solar wind parameters and emission heights is done by comparing simultaneous

one-minute values. This is OK for solar wind speed, which has a long autocorrelation time, but is questionable for IMF Bz, which can chance its polarity quite rapidly. This may be also significant for the conclusions drawn from the statistical results. I would like to see similar figures as in the current manuscript but using hourly means (rather than 1-minute means) of solar wind parameters in binning. This is a commonly used averaging in solar wind-magnetosphere coupling studies. See, e.g., Borovsky (2013) https://doi.org/10.1002/jgra.50110

*This is correct. It is the 1-minute values that have been compared in this study. There would be many ways to deal with the time delay in the comparison. We decided to go for the simplest, no averaged way. IMF variability is surely a good reason to average IMF data for the comparison. Below are the IMF Bz plots for Lapland without averaging (as in the original manuscript) and with 1h-averaged IMF data (as suggested).*

[Figure]

[Figure]

In the 1h-averaged version the averaged IMF Bz is required to be negative or positive respectively. The nightside height results from about 20MLT to about 03 MLT do not change. On the dusk side before 20 MLT the height distribution during negative IMF changes very little while that during the positive IMF becomes more variable when the IMF values are being averaged. Rather than physics the height fluctuations in the 1h-averaged version are due to the fact that during positive IMF conditions the number of data points becomes less evenly distributed between the bins. The number of data points is generally lower during positive IMF than during negative IMF, as well as towards dusk and dawn than around the midnight. On the dawnside the difference between positive and negative IMF shows similar

behaviour with and without averaging. Here, the averaging actually distributes the data points more evenly and the end result is smoother and clearer.

Same set of figures are shown below for Svalbard: original and 1h-averaged. Again the data points of the height distribution during positive IMF become a bit less evenly distributed between the bin when we average the IMF Bz, but the differences between the two plots are very minor. If something was to be brought up, it was the larger difference between positive and negative IMF in the post-midnight sector when averaged IMF has been used.

[Figure]

[Figure]

*Both figure pairs suggest that averaging describes the dawn sector precipitation better but matters less for the auroral precipitation energy in the other sectors. In the essence, the differences between the versions are subtle enough that we would rather describe in the text that this has also been tested without adding new figures or changing the current ones.*

I think the authors should cite papers showing that solar wind speed (and more accurately high-speed solar wind streams and the embedded Alfvenic Bz fluctuations) dominates the occurrence of substorms. The fact that different emission height distributions are found for low and high solar wind speed probably reflects the fact that

substorm activity is frequent during fast solar wind, but less frequent during slow solar wind.

E.g., Tanskanen et al., 2005, https://doi.org/10.1029/2005GL023318

*This is a very likely scenario indeed. The suggested sample reference and the substorm occurrence rate paper (Tanskanen, 2009) have been added to the revised version of the manuscript. However, we think that it may be unnecessary to discuss the Alfvenic fluctuations as the solar wind driving process but would rather leave it to making the connection between high-speed solar wind and high substorm occurrence, high substorm intensity and high substorm energy dissipation rate.*